# Skull-Base Chondrosarcoma: A Systematic Review of the Role of Postoperative Radiotherapy

**DOI:** 10.3390/cancers16050856

**Published:** 2024-02-21

**Authors:** Pawan Kishore Ravindran, Max E. Keizer, Henricus (Dirk) P. M. Kunst, Inge Compter, Jasper Van Aalst, Daniëlle B. P. Eekers, Yasin Temel

**Affiliations:** 1Department of Neurosurgery, Maastricht University Medical Center+, 6229 HX Maastricht, The Netherlands; max.keizer@mumc.nl (M.E.K.); j.van.aalst@mumc.nl (J.V.A.); y.temel@mumc.nl (Y.T.); 2Dutch Academic Alliance Skull Base Pathology, Maastricht University Medical Center+, Radboud University Medical Center, 6229 HX Maastricht, The Netherlandsinge.compter@maastro.nl (I.C.); danielle.eekers@maastro.nl (D.B.P.E.); 3Department of Otorhinolaryngology, Head & Neck Surgery, Maastricht University Medical Center+, 6229 HX Maastricht, The Netherlands; 4Department of Otorhinolaryngology, Head & Neck Surgery, Radboud University Medical Center, 6525 GA Nijmegen, The Netherlands; 5Department of Radiation Oncology (Maastro), GROW School for Oncology and Reproduction, Maastricht University Medical Center+, 6229 ET Maastricht, The Netherlands

**Keywords:** skull base, chondrosarcoma, low grade, radiotherapy, surgery

## Abstract

**Simple Summary:**

Currently, there is no global consensus regarding the treatment of skull-base chondrosarcoma. Some believe surgery is sufficient, while others administer adjuvant radiotherapy regardless of the extent of surgical resection. This review aims to gather the latest available evidence regarding the treatment of skull-base chondrosarcoma and to analyze the long-term prognosis to ascertain the potential added value of adjuvant radiotherapy and its application.

**Abstract:**

Surgery and radiotherapy are key elements to the treatment of skull-base chondrosarcomas; however, there is currently no consensus regarding whether or not adjuvant radiotherapy has to be administered. This study searched the EMBASE, Cochrane, and PubMed databases for clinical studies evaluating the long-term prognosis of surgery with or without adjuvant radiotherapy. After reviewing the search results, a total of 22 articles were selected for this review. A total of 1388 patients were included in this cohort, of which 186 received surgery only. With mean follow-up periods ranging from 39.1 to 86 months, surgical treatment provided progression-free survival (PFS) rates ranging from 83.7 to 92.9% at 3 years, 60.0 to 92.9% at 5 years, and 58.2 to 64.0% at 10 years. Postoperative radiotherapy provides PFS rates ranging between 87 and 96.2% at 3 years, 57.1 and 100% at 5 years, and 67 and 100% at 10 years. Recurrence rates varied from 5.3% to 39.0% in the surgery-only approach and between 1.5% and 42.90% for the postoperative radiotherapy group. When considering prognostic variables, higher age, brainstem/optic apparatus compression, and larger tumor volume prior to radiotherapy were found to be significant factors for local recurrence.

## 1. Introduction

Chondrosarcomas are a prominent group of sarcomas, second to osteosarcomas in frequency. The term ‘chondrosarcoma’ encapsulates a diverse group of malignant tumors stemming from chondroid cells in the appendicular and axial skeleton. It accounts for approximately 11% of primary malignant bone tumors, with an estimated incidence of 1 in 200,000 in the United States [1,2]. Cases of chondrosarcoma primarily affect the cartilaginous cells of the arms, shoulders, legs, spine, and pelvis. Chondrosarcomas of the skull base account for approximately 1% of these cases and mostly arise de novo [2]. Inversely, skull-base chondrosarcomas account for approximately 6% of all skull-base tumors and 0.15% of all intracranial neoplasms [3,4].

Current literature suggests that most primary skull-base chondrosarcomas occur in the middle fossa, namely at the petrous bone, the clivus, and/or the petroclival synchondrosis [2]. Macroscopically, chondrosarcomas are often bigger than 2 cm in diameter and are characterized by being hard, smooth, and lobulated [3]. Chondrosarcomas typically have (1) an abundant hyaline-type cartilaginous stroma and (2) neoplastic chondrocytes. There also may be myxoid or mucoid material and cystic changes seen macroscopically. Malignancy is characterized by the infiltration into the bony trabeculae [4]. Microscopically, chondrosarcomas house irregularly shaped cartilage lobules separated by fibrous bands. The chondrocytes are usually atypical and variable in size. Myxoid changes and matrix liquefaction can also be seen microscopically [5].

Histological analysis categorizes chondrosarcomas into four subgroups: conventional, mesenchymal, clear cell, and dedifferentiated chondrosarcoma. Additionally, chondrosarcomas can be grouped into three grades as per the cell differentiation—as introduced by the World Health Organization: Grade I (well-differentiated), Grade II (intermediately differentiated), and Grade III (poorly differentiated) [4]. Grade I is characterized by low cellularity, low to no myxoid, and abundant chondroid matrix. A conventional chondrosarcoma would, for example, fall in this category of Grade I. Grade II is recognized by its increased cellularity, low chondroid matrix, and more prominent myxoid [6]. Examples of histological subtypes that fall into this grade include conventional and myxoid chondrosarcoma [6]. Lastly, Grade III is identified by its high cellularity, the presence of nuclear pleomorphism, no chondroid matrix, and myxoid stroma. Examples of histological subtypes that fall into this grade include conventional, mesenchymal, and dedifferentiated [6]. Conventional chondrosarcomas are the most common histopathological subtype; likewise, Grade I comprise the majority and, less commonly, Grade II [3]. Histologically, the clear-cell and conventional tumor types are associated with the highest overall survival, whereas the dedifferentiated and mesenchymal types were found to be associated with the lowest overall survival [7]. Similarly, Grade I and II tumors are associated with a better prognosis than Grade III [7].

Patients usually present at a median age of 42.5 years with symptoms dependent on the tumor size and location [2]. Aside from headaches, mostly located in the occipital or retro-orbital region, neuro-ophthalmological symptoms are the most reported signs [4]. Visual symptoms include blurred vision or loss of vision, ptosis, diplopia, and visual field defects. Other signs are often consistent with the consequences of brainstem compression and cranial nerve palsy [4]. Furthermore, this tumor can have endocrinological implications leading to hypopituitarism and diabetes insipidus because of the involvement of the sellar and/or suprasellar structures [4]. Bigger tumors may compress the brainstem.

The role of chemotherapy is limited in chondrosarcoma; thus, surgery and radiation therapy remain the two keystones for the treatment of chondrosarcomas [8]. Chemotherapy, however, is a viable option in locally advanced diseases or metastases [8]. Surgery is conducted for two reasons: to obtain tissue for diagnosis and to reduce tumor volume. The open, transcranial, transfacial and transpetrous, infratemporal, and endoscopic/microscopic endonasal approaches are all commonly used and efficacious options for surgical treatment of chondrosarcomas [3]. Although complete resection is favored, the anatomical constraints and proximity to critical neurovascular structures often make this an unrealistic philosophy. Despite these tumors being slow-growing, they show a high local recurrence, with metastasis occurring in approximately 10% of cases [3]. Common complications of surgery include cerebrospinal fluid leak, transient neurological complications such as cranial nerve neuropathies, meningitis, brain abscesses, and persistent neurological complications due to, for instance, an intracranial hemorrhage. Adjuvant radiotherapy is often given to treat the tumor bed because of postoperative micro/macroscopic remnants [3,4,7]. Commonly used adjuvant radiotherapy includes linear accelerator (LINAC)-based intensity-modulated radiotherapy (IMRT), stereotactic radiotherapy (SRT) with LINAC or Gamma Knife, and CyberKnife all using photons and particle radiotherapy (using, e.g., protons or carbon ions) [3,4]. Complications are associated with damage to the surrounding ‘organs at risk’ (OARs).

Despite these treatment options being available, each with its own benefits and caveats, no randomized clinical trials or large prospective series exist to define the ideal treatment. Furthermore, there are no studies comparing radiotherapy modalities (e.g., particles versus photons). This systematic review aims to discuss the current literature and describe the effect of adjuvant radiotherapy on long-term prognosis in patients with skull-base chondrosarcoma.

## 2. Methods

### 2.1. Search Strategy and Study Eligibility

This systematic review was performed in accordance with the Preferred Reporting Items for Systematic Reviews and Meta-Analyses (PRISMA) guidelines. PubMed, EMBASE (OVID), and Cochrane databases were searched using the following search input: (((skull base) OR (intracranial)) AND ((chondrosarcoma) OR (skull-base chondrosarcoma)) AND ((proton therapy) OR (radiotherapy) OR (proton beam therapy) OR (Gamma Knife) OR (carbon-ion therapy) OR (surgery) OR (photon therapy)) AND ((local control) OR (survival) OR (progression) OR (prognosis))). Studies published after 1999 were screened for inclusion; no other limits/filters were used. The last search was performed on January 5, 2024. Three authors (P.K.R., M.K., and Y.T.) went through the search results to identify clinical trials and observational studies conforming with our inclusion criteria. This was accomplished by first exporting the database searches to EndNote (Clarivate Analytics), after which duplicates were removed. Thereafter, one author (P.K.R.) independently (1) screened the title and abstracts and excluded the articles not suitable to the goal of this review; and (2) of the remaining articles, the full text was analyzed and was included if eligible. This was confirmed by two other authors (M.K. and Y.T.). If the full article was not available, the authors were contacted to obtain access. Disagreements between authors were resolved by consensus. This review protocol was registered on PROSPERO, ID: CRD42024496554.

Inclusion and exclusion criteria were defined prior to the search. Studies were included if (1) they were peer-reviewed original articles about patients with skull-base chondrosarcomas who have undergone surgery alone or received postoperative adjuvant radiation therapy, (2) reporting the prognostic outcomes (e.g., overall survival, local control, and disease-specific survival), and (3) they were written in English. Studies were excluded if they were (1) literature or systematic reviews, meta-analyses, case reports, comments, books, information pages, animal or phantom studies, (2) histological studies, and (3) written in a foreign language.

### 2.2. Data Extraction and Quality Assessment

Data were extracted from the search inputs by one author (P.K.R.) and confirmed independently by two additional authors (Y.T. and M.K.). The extracted data included the following: (1) study information, including research design, authors, and year of publishing; (2) characteristics of patients, including sex, mean age, mean follow-up time, tumor size, tumor pathology/grade type of intervention, the surgical approach, mean dose of radiotherapy, type of radiotherapy, and whether they were treated at initial diagnosis or at recurrence; and (3) outcomes, including progression-free survival (PFS), overall survival rates (OS), disease-free survival (DFS), complications or adverse events, and recurrence rates. In this paper, the distinction was made between low-grade, including Grades I and II, and high-grade chondrosarcomas, including Grade III. For simplification purposes, local control and recurrence-free survival were considered identical to progression-free survival. Thus, local control was used in studies where progression-free survival was not available, and when local control was not available, recurrence-free survival was used if provided. The scale used to measure acute and late adverse events is mentioned in the results if reported by the original study. The study authors were contacted once when the studies did not report enough data to compute the effect size.

### 2.3. Reporting Bias Assessment

The quality of the included studies was primarily assessed using the Cochrane Risk of Bias in Nonrandomized Studies- of Interventions (ROBINS-I) tool [9]. For studies that used a format conforming to a case series, as opposed to a cohort study, the JBI quality assessment tool for case series was used [10]. These tools assess the confounding, selection bias, classification of interventions, deviations from intended interventions, missing data, measurement of outcomes, and reporting bias. The studies assessed using the ROBINS-I were assessed on a scale of risk: low risk, moderate risk, serious risk, critical risk, or no information.

### 2.4. Statistical Analysis

The primary outcome of this review was to assess the prognosis (i.e., OS, PFS, DFS, and LC) of the intervention, i.e., surgery with or without postoperative radiotherapy. The data on patient characteristics and clinical information from all the studies were pooled to obtain the mean baseline characteristics. This was performed using SPSS V.28 (IBM SPSS Statistics, IBM Corp, Armonk, New York, NY, USA). After consulting with a professor in statistics, it was deemed impossible to pool the data from the survival of all the studies as the majority of the studies did not report the number at risk of the Kaplan–Meier curve, the effect size (i.e., hazard ratio, mean difference, standard error), or the confidence intervals. As a result of this heterogeneity, we resorted to reporting a systematic review without a meta-analysis.

## 3. Results

### 3.1. Literature Search

The literature search identified 830 results from PubMed, EMBASE (OVID), and Cochrane Library, of which 56 were selected based on the study designs, titles, and abstracts (see Figure 1). Further assessment of the full texts resulted in 34 papers being excluded, and eventually, 22 papers were included in this review; more information is shown in Figure 1.

### 3.2. Risk of Bias Assessment

As per the ROBINS-I tool, the following confounding domains were identified in the protocol phase of this study: age, sex, extent of resection, tumor size, tumor pathology/grade, radiotherapy dose, and type of radiotherapy. No co-interventions were determined to be relevant for this risk of bias assessment. Studies of a case-series design were assessed using the JBI’s quality measurement tool for case series. The risk of bias assessment of all articles ranged from moderate risk to critical risk, portrayed in Appendix A.

### 3.3. Overview of Results

Table 1 shows the pooled characteristics of the patients from all the included studies with regard to sex, mean age, presenting symptoms, whether treatment was given at initial diagnosis or recurrence, tumor location, surgical approach, and presenting cranial nerve palsies. To summarize, this review includes a total of 1388 patients, with a mean age of 41.81 years. There were 631 (45.5%) male and 757 female (54.5%) participants in this review. The median age was approximated as being the mean because, in studies where both median and mean age were reported, little difference was present between the two. Patients typically harbored low-grade chondrosarcoma (98.9%), located primarily in the petroclival area. They were primarily treated at initial diagnosis and usually presented with diplopia, headaches, and visual field/hearing deficits. When considering cranial nerve palsies, abducens palsy is most common, possibly accounting for the diplopia.

The included studies were conducted from across the world, primarily at specialized hospitals. Of the 22 articles, all were retrospective in design, and the majority of them were single-arm cohort studies. Tzortzidis et al. [11], Simon et al. [12], and Hasegawa et al. [13] were the only studies to set up a two-arm retrospective cohort study comparing the surgery-only approach to giving postoperative radiotherapy. Overall, the surgical approaches used to remove these tumors have developed over the years, as acknowledged by several of the included studies; when pooling the reported approaches, we find that the endonasal, subtemporal–infratemporal, and pteronial approaches are the most popular. This reflects the most suitable technique for the patients as well as the familiarity of the technique for the operating surgeons. As no meta-analysis was conducted, we present the ranges of the outcomes of the patients in each cohort. Table 2 shows the characteristics of the included studies with respect to sample size, intervention, extent of resection, grade of chondrosarcoma, tumor size, mean dose, follow-up time, progression-free survival, overall survival, and disease-specific survival.

### 3.4. Surgery Only

Out of the 22 included studies, 7 investigated the long-term prognosis of patients treated with surgery only. A total of 186 patients underwent surgery, with salvage therapy upon recurrence. Excluding the study by Tzortzidis et al. [11] (due to a lack of baseline characteristics specific to the surgery-only group), there were 72 (45.9%) males and 85 (54.1%) females. The average tumor size in this cohort was 35.94 cm^3^. The tumors resected were primarily low-grade (99.4%); the extent of resection is shown in Table 2. The majority of patients were treated at the initial diagnosis (66.3%). With mean follow-up periods ranging from 39.1 to 86 months, surgical treatment provided a progression-free survival (PFS) ranging from 83.7 to 92.9% at 3 years, 60.0 to 92.9% at 5 years, and 58.2 to 64.0% at 10 years. Similarly, when considering the overall survival (OS), we find it ranging from 68.0 to 95.0% at 5 years, based on two studies. As no more than one study was available at the 3- and 10-year marks, no range of overall survival could be provided at these time points. As the surgical aim and technique play a vital role in the long-term prognostic outcomes, we shall highlight the key surgical features and reconstruction techniques from each study in this cohort.

The surgical intention of Tzortzidis et al. [11] was to obtain a complete tumor resection. A second, or rarely third, operation was staged and conducted with extensive tumors or when large tumor residue was seen on postoperative MRI. Radiotherapy was given when patients underwent subtotal resection. Different surgical approaches were used but were not explicitly mentioned. Internal carotid artery resection and grafting were necessary in six cases. Various vascularized flaps were used for reconstruction, including the temporalis muscle flap, pericranial flap, and rectus abdominis free flap supplemented by free-fat grafts.

In Samii et al. [14], the operative approaches were chosen on the basis of the predominant direction of tumor extension. When there was brainstem and/or optic apparatus compression, the primary goal was to relieve this pressure and only secondarily excise the tumor. The most common approaches were retrosigmoid (in 12 surgeries), pteronial approach (in 9 surgeries), and transethmoid approach (in 6 surgeries). Using the retrosigmoid approach, total tumor removal was achieved in 58% of cases, compared to 33% with the other two techniques. In case of dural penetration/opening, a fat graft was used, and a lumbar drain was placed to reduce the chance of CSF leak.

In the study by Vaz-Guimaraes et al. [15], the patients underwent endoscopic endonasal surgery. The aim of the surgery was the complete removal of the tumor, except for three patients (due to extensive disease and poor condition). For extensive chondrosarcomas, staged and combined operations were used. The surgical approach used was determined by tumor location and included the anterolateral approach, transpetrous and far-lateral approach, and transcervical approach. Intradural tumor resection, dural opening, or extensive skull-base drilling was followed by a multilayer reconstruction. The reconstruction techniques varied, including the use of pedicled vascularized flaps (e.g., the pedicled nasoseptal flap) and inferior turbinate, pericranial, and temporoparietal fascial flaps. If the dura was not opened or when limited cranial base drilling was performed, reconstruction was conducted with tissue glue only.

The patients in the study by Hasegawa et al. [16] all underwent endoscopic transnasal surgery for their skull-base chondrosarcomas. Tumors were approached either via the transsphenoidal route, transpterygoid approach, or transmaxillary approach. Tumors located in the lower clivus, craniovertebral junction, and the nasopharynx were approached via the upper pharynx. Tumors in the anterior skull base were approached directly through the ethmoidal sinus. The aim was to attain maximal retention—in the case of remnant tumor components—adjuvant SRS was advised at 3–6 months after surgery. The reconstruction technique was not reported. Simon et al. [12] report limited data regarding the surgical philosophy. The surgical approach included pterional, infratemporal fossa, retrosigmoid, lateral rhinotomy, and endonasal in the surgery-only group.

The study by Al Shaibibi et al. [17] adopted a surgical aim to safely resect as much of the tumor as possible whilst preserving the internal carotid artery and the cranial nerves V, VI, VII, and VIII. All patients in this study were operated on using an epidural anterior petrosectomy approach. The surgical resection began with identifying the upper pole of the tumor, whereafter, piecemeal resection of the tumor ensued; this was followed by the use of ring curettes to allow for additional resection. Parts of the dura were opened and resected if the tumor mass infiltrated the meningeal layers despite the tumors all being epidural. The petrous cavity was filled with abdominal fat, and the temporal bone flap was secured with titanium plates and screws.

Liu et al. [18] included patients whose tumors were removed by craniotomy (n = 20) and by endoscopic endonasal transsphenoidal approach (n = 4). The craniotomy approach included the pterional or expanded pterional approach, posterior sigmoid sinus approach, orbitozygomatic approach, paramedian approach, posterior median extension approach, and combined supratentorial and infratentorial approach. The reconstruction technique and surgical aim were not reported.

### 3.5. Radiotherapy

Seventeen studies report the long-term prognosis of patients treated with postoperative radiotherapy. A total of 1202 patients underwent postoperative radiotherapy, of which 530 (44.8%) were male and 654 (55.2%) were female, excluding Tzortzidis et al. for the aforementioned reason [11]. The extent of resection is reported, per study, in Table 2. The tumors in this cohort were primarily low-grade (98.9%), and the majority were treated at initial diagnosis (78.0%). The mean/median tumor volume and radiation dose are also provided in Table 2. This review reports several forms of radiotherapy: namely, proton therapy (n = 765), carbon-ion therapy (n = 212), a combination of proton and photon therapy (n = 149), photon therapy (n = 18), and Gamma Knife radiosurgery (n = 58). Patients who received subtotal resection in the study by Tzortzidis et al. [11] also received postoperative radiotherapy. It should be noted that the long-term recurrence-free survival of these patients seems like an outlier (e.g., a 10-year recurrence-free survival of 13.8%). Other patients in this cohort presented with higher long-term prognostic outcomes despite having received subtotal resections prior to radiotherapy. This discrepancy may be explained due to a possible delayed time between the subtotal resection and subsequent radiotherapy. Furthermore, there was no standardization in the radiotherapy received: proton therapy was given to 15.6%, radiosurgery (Gamma Knife or CyberKnife) to 68%, and fractionated radiation to 15.6%. Due to these discrepancies, we have excluded this study in consideration when providing the ranges of PFS and OS.

The follow-up period reported in each study is shown in Table 2. Postoperative radiotherapy provides PFS rates ranging between 87.0 and 96.2% at 3 years, 57.1 and 100% at 5 years, and 67.0 and 100% at 10 years. As for the OS, the following ranges are seen: 87.8–100% at 3 years, 84.1–100% at 5 years, and 78.1–87% at 10 years. Specific to proton therapy, the following PFS ranges are seen: 90–100% at 2 years, 75–100% at 5 years, and 84.2–100% at 10 years. The results are similar to the overall OS when analyzing the OS of patients undergoing proton therapy at the aforementioned time points, as the majority of the studies included in this review investigate the efficacy of proton therapy. Carbon-ion radiotherapy was evaluated by three studies [19,20,21], from which a PFS between 95.9% and 97.2% and an OS between 96.1% and 98.5% were seen at 3 years. Only Sahgal et al. [25] investigated the long-term prognosis of patients treated with image-guided intensity-modulated photon therapy. Two studies reported the long-term prognosis of postoperative GKRS [31,32]; however, they contain methodological differences as to when the GKRS was given and how the PFS was measured.

### 3.6. Recurrence and Prognosis

The recurrences, salvage treatment, and prognostic factors reported from the included studies are depicted in Table 3. Unless explicitly reported, the percentages given in this table were calculated based on the number of recurrences divided by the total sample size. In the articles assessing the surgery-only approach, a tumor recurrence rate between 5.3% and 39.0% was seen. In these cases, salvage therapy was given, which primarily consisted of surgery ± radiotherapy (commonly proton therapy or Gamma Knife radiosurgery), which provided sufficient tumor control in most cases. As for the radiotherapy group, the recurrence rate ranged between 1.5% and 42.90%. As the study by Tzortzidis et al. faced high loss to follow-up (>50%) [11], its recurrence rates were not included in the aforementioned ranges.

### 3.7. Complications

The patients of the selected articles were pooled together, and a summary of the surgical complications, acute and late radiation-induced adverse effects is reported in Table 4, Table 5 and Table 6, respectively. There were 59 reported deaths due to the tumor and 36 deaths due to intercurrent disease among the selected articles. Radiation-induced adverse effects are split into acute and late adverse effects. Articles that did not grade the complications as per the CTCAE classification [33] were listed as ‘N.M’: not mentioned.

## 4. Discussion

Due to the rarity of skull-base chondrosarcomas, the number of studies investigating the efficacy of therapeutic options has been limited. This systematic review included 22 studies investigating the long-term efficacy of surgical therapy and postoperative radiotherapy, with a total of 1388 patients. The majority of the skull-base chondrosarcomas were low-grade, and there was no predilection for the sex of the patient. This review presents a rather broad range of the long-term outcomes of survival primarily due to the heterogeneity in the data included and the lack of detail in the Kaplan–Meier survival analysis provided in each study. The majority of the articles assess the patients up to the 5-year mark, most up to the 10-year mark, and seldom to 15 years.

The treatment of skull-base chondrosarcoma requires a multidisciplinary approach, with the involvement of the neurosurgeon, ENT surgeon, neuro-radiologist, neuropathologist, radiation therapist, and medical oncologist. Surgical resection of the chondrosarcoma is considered the first-line therapy; however, it also provides the opportunity to confirm the diagnosis at the histological and molecular level. Radical resection is often not possible, and instead, a maximal, safe resection approach is undertaken to spare the surrounding critical structures and decrease iatrogenic morbidity. The overall prognosis in the surgery-only group ranges widely, especially as the follow-up time increases. However, one constant observation is the steep decline in the lower limit of the prognostic ranges, from year 1 to year 10, in the surgery-only group (−25.5%). This is in contrast to the postoperative radiotherapy group, which does not see a steady decline in the lower limit (−20.0%). As the number at risk is not reported, it is difficult to ascertain whether this lack of a steady decline in the lower limit should be attributed to a loss of follow-up. Loss of follow-up in a particular subset of patients who underwent postoperative radiotherapy (e.g., those that showed better recovery) would shift the composition of the sample, thereby also resulting in an altered prognosis. Furthermore, the upper limit of the ranges remains constant in the postoperative radiotherapy group, whilst declining in the surgery only group (−28.9%). Although only two studies reported the ten-year outcome of patients treated with surgery only in this review, it is notable that the upper limit of the ranges show a greater decline than its lower limit at 10 years. These results would suggest a better long-term control with adjuvant radiotherapy; however, this was not reflected in a systematic review conducted in 2009, where the surgery-only group had better 5-year prognostic outcomes [34]. This discrepancy might possibly be explained by the fact that newer studies became available, further clarifying the true prognostic differences between the two intervention groups.

When analyzing the surgery-only group, it is important to recognize that data were extracted from only seven studies. The majority of the patients in this group underwent a gross total resection (61.2%), with patients undergoing multiple staged surgeries if needed. Furthermore, the most common surgical approaches were endonasal, subtemporal–infratemporal, and pterional. The endonasal approach has taken the spotlight in recent literature, showing good prognostic outcomes. Endoscopic surgery provides a better route to the tumor than the transcranial approach, considering most originate in the petroclival area. As reported by Schwartz et al., the transcranial approach is more likely to traverse cranial nerves than the endoscopic approach. The gross total resection rates are reported to be around 66.7–80.0% with good tumor control as opposed to a GTR rate of 50–100% in the transcranial approach [35]. Moreover, endoscopic nasal approaches allow for shorter hospitalization, decreased complications, and decreased mortality [36]. With the development of new approaches, such as the transmaxillary approach, the use of endoscopic skull base surgery is only increasing. The reconstruction method, reported in four of the eight studies, was accomplished using vascularized flaps or fat grafts. The decision between the use of abdominal fat or vascularized flaps for reconstruction is fueled by the indication alongside surgical expertise [37]. Complications in skull base surgery for tumor resection reported in the literature include intracranial bleeding, blindness, CSF leak, osteonecrosis, cerebral abscess, meningitis, cranial nerve neuropathies, and cosmetic deformities [36]. Many overlap with the complications seen in this review. Studies indicate that CSF leak rates are reduced when the reconstruction is performed with vascularized local tissue, particularly in patients with a history of radiotherapy [36]. Furthermore, multilayered reconstruction is reported to be superior to single layer [36]. There is more to explore regarding the skull base reconstruction techniques; however, this falls beyond the scope of this paper.

When considering the postoperative radiotherapy group, 17 studies investigated the efficacy of this intervention, with proton therapy being the most commonly investigated. Contrary to the surgery-only group, only 6.3% of the reported cases were gross total resections. The remaining patients underwent either a partial resection (44.8%), subtotal resection, or biopsy (4.9%) prior to irradiation. From the studies included, proton and carbon therapy provide better tumor control than Gamma Knife radiosurgery and photon radiation therapy. These promising results for proton therapy obtained in this review are also reflected in the current literature. For example, in the review by Noel et al. assessing the role of proton therapy in various intracranial tumors, an overall local control between 85 and 100% is found at 3 years and between 75 and 95% at 5 years for chondrosarcomas [38]. Similarly, the carbon-ion group, despite only having three studies in this review, shows promise with high control rates. These results are further confirmed in the systematic review by Holtzman et al., which includes the articles selected for this review, from which 1-, 3-, and 5-year control rates of 99%, 89%, and 88%, respectively, were reported [39]. We see a significant improvement in the long-term prognosis when administering postoperative proton therapy, with the lowest reported PFS being 75% at 5 years, as opposed to 64.0% with surgery only at 5 years. The literature review by Kano et al. provides a better overview of the long-term efficacy of Gamma Knife SRS for skull-base chondrosarcoma [40]. The review compiles 46 patients from five studies, of which one is included in this review, who underwent postoperative GKRS. Without prior radiotherapy, the PFS rates were 92%, 88%, and 81% at 1, 3, and 5 years, respectively [40]. Although not as effective as particle therapy, it provides the clinician with an alternative option for primary or salvage treatment.

Despite these differences, there are similarities between the radiotherapy groups, most notably in the overall survival, which was similar for both carbon-ion and proton therapy cohorts. We believe that this arises from their similar particle characteristics. A particular advantage that particle therapy offers is a steeper dose gradient to the organs at risk adjacent to the gross tumor volume compared to non-particle radiotherapy. This is related to the Bragg peak and curve, which depicts the deposition of energy as the particle travels through matter [41]. Promising control of the Bragg curve is shown with the pencil beam system. Carbon-ion therapy has a higher biological effectiveness and is associated with more side effects than proton therapy [41]; therefore, this similarity in overall survival might change when investigated with a greater sample size and a longer time frame. Only Sahgal et al. investigated the efficacy of image-guided intensity-modulated (IG-IMRT) photon therapy. It presents a lower control rate and overall survival than the particle radiotherapy options. The advantage of IMRT is due to its inverse treatment planning system, which allows high-dose conformation to reach the target volume whilst reducing the dose to surrounding non-target tissues [42]. Traditionally, particle therapy was used in the treatment of skull-base chondrosarcoma. Hence, due to the lack of studies and the heterogeneity in available studies, it is difficult to compare photon radiation with particle therapy and to draw sound conclusions. The review by Fossati et al. reports subpar results compared to particle therapy, possibly accounted for by the lack of adequate target coverage and radiation dosage [42].

Although radiotherapy offers an improvement in the treatment of skull base cancers, it also comes with complications. The complications listed in Table 5 overlap with the ones reported in the current literature [36]. The same applies to the reported late radiation-induced adverse events in this review, with literature reporting pituitary gland insufficiency, ocular pathway damage, sensorineural hearing loss, and temporal lobe necrosis being the most common complications [1]. These complications are of utmost importance to consider due to the close proximity of critical organs to the tumor location. The rate of complications was far higher in the studies investigating the efficacy of adjuvant radiotherapy. We expect two possible explanations for this difference: (1) this reflects the true difference in the complication rate between the surgery-only approach and giving postoperative radiotherapy, or (2) the surgical complications are underreported, as not all studies did so. However, findings consistent with the former explanation were found in the review by Palmisciano et al. [2].

Although the range of the rate of recurrences provided was broad, it is evident that the recurrences were more frequent in the surgery-only group. This can be elucidated when the upper limit of the recurrence rate in the radiotherapy group is properly evaluated. Due to the small sample size in the study by Pattankar et al., which only included seven patients, any recurrence was bound to be high in percentage. Realistically, when considering this study as an outlier, one obtains a recurrence range between 1.5% and 18.4% in the postoperative radiotherapy group. Similar results are seen in the study by Takahashi et al. [43], which reports the recurrence rates of skull-base chondrosarcoma and chordoma in patients treated with proton beam therapy. In this study, recurrence rates of 0%, 0%, 0%, 0%, and 7.1% were reported for skull-base chondrosarcoma patients at 1, 2, 3, 4, and 5 years, respectively [43]. When considering the prognostic variables, the risk factors that increase the likelihood of poor tumor control and overall survival include a large tumor volume, brainstem/optic apparatus involvement, older age, and an increased number of prior surgeries. These factors are also reported in previous literature, for example, in Kremenevski et al., although most reviews combine data from chordoma and chondrosarcoma studies [4].

There are certain limitations to consider for this review. Although the search input was well-generated, abiding by the PICOT structure, the articles obtained were dubious in terms of bias. All were retrospective observational studies dating back to 1999. Several articles did not account for potential confounding factors specific to their analysis. As a result, the risk of the included articles ranges from moderate to critical bias as per the ROBINS-I tool. Furthermore, the development of surgical and radiotherapy techniques/protocols between 1999 and current times is an important consideration when analyzing the reported prognostic outcomes in older studies. Furthermore, the included articles do not further detail their Kaplan–Meier curves, restricting us from creating a meta-analysis. This limits this review to be merely descriptive in nature and restrains us from providing a statistical answer to prove the superiority of one intervention over the other. Furthermore, some studies included were limited by the small sample size; in these cases, the percentages reported might simply be inflated. An open-access database where the prognostic outcomes alongside baseline data of patients with skull-base chondrosarcoma are included seems ideal to counter this limitation, especially considering its rarity and as it would allow for robust statistical analysis. Another limitation that this review faced was the short follow-up period of maximally 10 years in most studies. Although not surprising considering the rarity of this condition. Additionally, several excluded and some included articles grouped chordoma and chondrosarcoma patients together, impairing our ability to extract data specific to the latter group of patients. Future research can perhaps elaborate on the prognostic outcomes at a longer follow-up period specific to the chondrosarcoma patient. Another limitation revolves around the prognostic outcomes used, as it only assesses the tumor control. Measuring patient-oriented outcomes, such as quality of life, in order to properly ascertain the impact of radiation related complications and long-term prognosis would provide a more patient-oriented answer to integration of radiotherapy into their care. Lastly, the majority of the patients were diagnosed with low-grade chondrosarcoma, therefore leaving the treatment of high-grade cases an unexplored terrain for future research.

Future research comparing the local control provided by different surgical techniques on a longer term would be beneficial. It would allow clinicians to understand the best operative route specific to tumor location whilst giving them an idea of the expected control rate. Furthermore, nuances in the endoscopic technique can further be explored, thereby further substantiating its efficacy as a surgical option. As for radiotherapy, we suggest further investigation of the different forms of radiation delivery, for example, investigating the long-term prognosis of patients treated with image-guided radiotherapy, intensity-modulated radiotherapy, or pencil beam scanning. Such techniques allow for better control of the dosage of radiation delivered to the tumor whilst sparing the nearby critical organs. Such research has been slowly introduced since 2016, with papers such as Sahgal et al. [17] and Weber et al. [21], and show promising results. Furthermore, assessing the use of hypofractionated versus hyperfractionated radiotherapy in terms of tumor control can be another point of focus. Recent studies, such as in Sallabanda et al., have shown promising results for a hypofractionated schedule for proton therapy in providing good tumor control [44].

Furthermore, investigating other forms of particle therapy than proton therapy is another potential focus. More specifically, this review has shown the potential for carbon-ion radiotherapy. Although the dose distribution of charged particles are similar, there exist discrepancies between different particles—as seen in carbon-ion vs. proton therapy [38]. With carbon ions, there is a sharper dose falloff at both the distal and lateral beam edges, essentially targeting the tumor in a more precise fashion [38]. Moreover, when analyzing the amount of radiation energy deposited along a particle’s trajectory in a particular medium (quantified by the linear energy transfer, LET), we find that carbon has a higher LET than protons. The high LET in carbon ions is theoretically more optimal for tumor control and also spares tissues upstream of the target [38]. Considering these potential advantages, we believe that carbon-ion radiotherapy might be an important option to investigate.

Further research can also focus on the logistics behind the treatment of patients with chondrosarcoma. For example, analyzing the cost-effectiveness of particle therapy and its implementation in healthcare.

## 5. Conclusions

Skull-base chondrosarcomas remain a challenge to treat due to their location and close proximity to vital structures. Surgery has been the primary treatment option chosen for patients harboring a skull-base chondrosarcoma, with recent literature supporting the use of an endoscopic approach. However, the recurrence rate is still high, and usually, some form of salvage therapy is used to treat for recurrence. This review shows promising results in terms of long-term prognosis and recurrence rate favoring the use of postoperative radiotherapy.

## Figures and Tables

**Figure 1 cancers-16-00856-f001:**
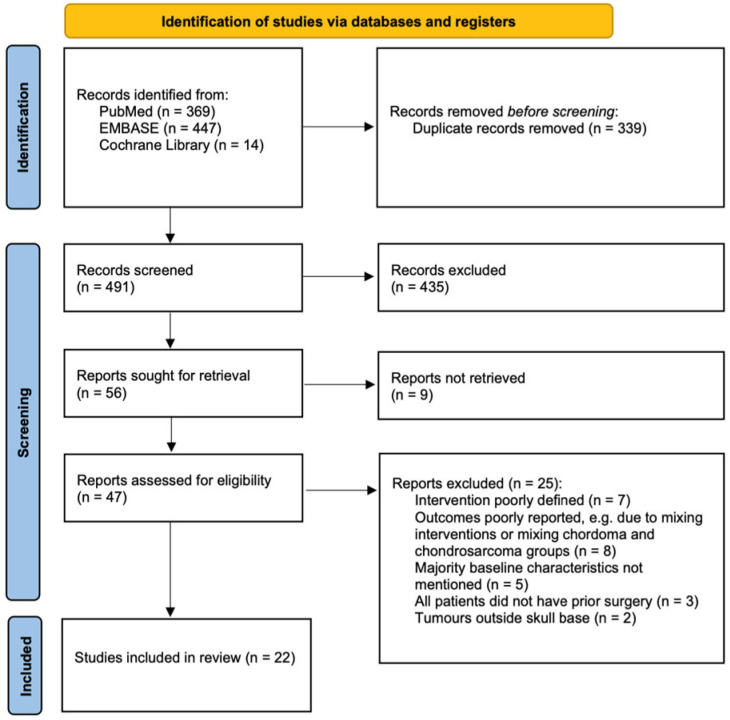
PRISMA flow diagram showing inclusion process.

**Table 1 cancers-16-00856-t001:** Pooled baseline patient characteristics.

Characteristic	Total Number of Patients
Sex	
Male	567 (45.3%)
Female	686 (54.7%)
Mean Age ± SD (range)	41.9 ± 4.1 (32–52)
Intervention	
Surgery only	186 (13.4%)
Postoperative radiotherapy	1202 (86.5%)
Proton therapy	765 (63.6%)
Carbon-ion therapy	212 (17.6%)
Proton and photon therapy	149 (12.4%)
Gamma Knife	58 (4.8%)
Photon	18 (1.4%)
Surgery at	
Initial diagnosis	67 (66.3%)
Recurrence	34 (33.7%)
Postoperative radiotherapy at	
Initial diagnosis	486 (78.0%)
Recurrence	137 (21.9%)
Presenting Symptoms	
Diplopia	157 (38.0%)
Headaches	77 (18.6%)
Hearing deficit	56 (13.6%)
Visual field deficit	23 (5.6%)
Nasal issues	20 (5.6%)
Dizziness	16 (3.9%)
Coordination problems	11 (2.7%)
Swallowing difficulties	7 (1.7%)
Facial weakness	7 (1.7%)
Facial pain	6 (1.4%)
Tinnitus	6 (1.4%)
Hemiparesis/amaurosis	5 (1.2%)
Miscellaneous	22 (5.3%)
Tumor Location	
Petrous bone	158 (15.1%)
Petroclival	144 (14.8%)
Clivus	142 (14.5%)
Tempero-occipital junction	137 (14.1%)
Cavernous sinus	123 (12.6%)
Sphenoid bone	45 (4.6%)
Middle cranial fossa	26 (2.7%)
Anterior skull base	25 (2.6%)
Suprasellar	22 (2.3%)
Sphenopetrosal	20 (2.1%)
Parasellar	19 (1.9%)
Ethmoid bone/paranasal sinus	14 (1.4%)
Posterior cranial fossa	12 (1.2%)
Posterior fossa	12 (1.2%)
Jugular fossa	11 (1.1%)
Miscellaneous	44 (4.5%)
Surgical Approach	
Endonasal	85 (31.2%)
Subtemporal–infratemporal	44 (16.5%)
Transcranial	24 (8.9%)
Pteronial	19 (7.1%)
Retrosigmoid	18 (6.7%)
Extended sub-frontal	18 (6.7%)
Frontotemporal-transzygomatic	11 (4.1%)
Epidural anterior petrosectomy	10 (3.7%)
Lateral rhinotomy	7 (2.6%)
Transethmoid	6 (2.2%)
Infratemporal fossa	3 (1.1%)
Extreme lateral transcondylar	3 (1.1%)
Subtemporal	3 (1.1%)
Transcranial-transnasal	3 (1.1%)
Miscellaneous	13 (4.8%)
Preoperative Cranial Nerve Palsies	
Abducens	202 (30.6%)
Trigeminal	70 (10.6%)
Facial	50 (7.6%)
Vestibulocochlear	46 (6.9%)
Oculomotor	44 (6.7%)
Hypoglossus	26 (3.9%)
Vagus	22 (3.3%)
Glossopharyngeal	20 (3.0%)
Optic	14 (2.1%)
Accessory	11 (1.7%)
Trochlear	11 (1.7%)
Olfactory	3 (0.5%)
Unmentioned	140 (21.2%)
Total Preoperative CN palsies	659

This table depicts the sex, mean age, intervention used, treatment at initial diagnosis or recurrence, baseline presenting symptoms, tumor location, surgical approach used, and lastly, the preoperative cranial nerve palsies. Frequencies of less than 1% were grouped into the respective categories as miscellaneous. In the presenting symptoms, examples of symptoms that fell into the miscellaneous group include fatigue, proptosis, and cognitive impairment. In tumor location, examples of locations that fell into the miscellaneous group include the orbital apex, infratemporal fossa, and cervical spine. Lastly, in the surgical approach group, examples of approaches that fell into the miscellaneous group include maxillectomy, presigmoid, and transoral transmandibular approaches.

**Table 2 cancers-16-00856-t002:** Raw data of selected articles. See Appendix A for quality assessment of Studies 4 and 21. GTR: gross tumor resection, STR: subtotal resection, NTR: near-total resection PR: partial resection, P.O. = postoperative, R.T = Radiotherapy, N.M: not mentioned, N/A = not applicable, y = years, n = number of patients.

Study Number	Author, Year	Number of Patients (No. Males)	Intervention	Extent of Resection (%)	Tumor Pathology	Tumor Size	Mean Dose	Follow Up	Progression-Free Survival	Overall Survival	Disease-Specific Survival
1	Tzortzidis et al., 2006 [11]	47 (24)	Surgery only (n = 29) Adjuvant RT (proton, radiosurgery, fractionated radiation) (n = 18)	GTR in 29 (62) STR in 18 (38)	Low grade, n = 45 High grade, n = 2	N.M	N.M	Mean: 86 months	GTR: 3y: 92% 5y: 78.3% 10y: 42.3% STR + RT: 3y: 78% 5y: 41% 10y: 13.8%	N.M	N.M
2	Simon et al., 2018 [12] Moderate Risk	Surgery only: 24 (10) Surgery + RT: 23 (13)	Surgery only (n = 24) P.O proton RT (n = 23)	Surgery only: GTR in 13 (54) PR in 11 (46) Surgery + RT: GTR in 3 (13) PR in 20 (87)	Low grade, n = 47	Mean (surgery only) = 39 cm^3^ Mean (P.O. proton therapy = 33 cm^3^	Mean: 70 GyE	Mean: 91 months	Surgery only: 5y: 67.8% 10y: 58.2% P.O proton radiotherapy: 5y: 100% 10y: 87.5%	N.M	Surgery only: 5y: 89.8% 10y: 89.8% P.O proton radiotherapy: 5y: 100% 10y: 100%
3	Hasegawa et al., 2021 [13] Moderate Risk	Surgery only: 18 (6) Surgery + RT: 14 (9)	Surgery + proton RT (n = 11) Surgery only (n = 18) P.O photon RT (n = 1) P.O photon + proton therapy (n =2)	Surgery only: GTR in 10 (56) Non-GTR in 8 (44) P.O RT: GTR in 1 (7) Non-GTR in 13 (93)	Low grade, n = 32	Mean diameter in Surgery + RT group = 35 mm Mean diameter in Surgery group = 34 mm	70 GyE	Surgery + RT, median: 44 months Surgery only, median: 154 months	Surgery + RT 5y, 10y: 100% 15y: 67% Surgery only: 5y, 10y, 15y: 64%	N.M	Overall DSS: 5y: 100% 10y, 15y: 95%
4	Samii et al., 2008 [14] Critical Risk	25 (18)	Surgery only (n = 25)	N.M	Low grade, n = 25	Mean = 16.9 cm^3^	N/A	Mean: 80 months	N.M	5y, 10y: 95%	N.M
5	Vaz-Guimaraes et al., 2017 [15] Moderate Risk	35 (14)	Surgery only (n = 35)	GTR in 22 (63) NTR in 11 (31) STR in 2 (6)	Low grade, n = 35	Mean = 31.9 cm^3^	N.M	Mean: 44.6 months Median: 44 months	3y: 83.7% 5y: 80.8%	N.M	3y: 91.1% 5y: 90.5%
6	Hasegawa et al., 2018 [16] Critical Risk	19 (10)	Surgery only (n = 19)	GTR in 15 (79) STR in 2 (10.5) PR in 2 (10.5)	Low grade, n = 18 High grade, n = 1	Mean = 14.5 cm^3^	N/A	Median: 47 months	3y, 5y: 92.9%	N.M	N.M
7	Al-Shaibibi et al., 2023 [17]	10 (4)	Surgery only (n = 10)	STR in 10 (100)	Low grade, n = 10	N.M	N/A	Median: 70 months	1y: 90% 2y: 80% 5y: 60%	N.M	N.M
8	Liu et al., 2023 [18] Moderate Risk	26 (10)	Surgery only (n = 26)	GTR in 14 (54) STR in 10 (38) PR in 2 (8)	Low grade, n = 26	Mean: 65.13 cm^3^	N/A	Mean: 39.12 months	N.M	1y: 100% 3y: 81.8% 5y: 68%	N.M
9	Schulz et al., 2007 [19] Critical risk	54 (27)	P.O. carbon-ion RT (n = 54)	N.M	Low grade, n = 54	Median: 20 cm^3^	Median: 60 CGE	Median: 33 months	3y: 96.2% 4y: 89.8%	3y, 4y: 98.2%	N.M
10	Uhl et al., 2014 [20] Moderate risk	79 (39)	P.O carbon RT (n = 79)	PR in 62 Biopsy in 17	Low grade, n = 78 High grade, n = 1	Mean boost volume = 60.5 cm^3^	Median: 60 GyE	Median: 91 months	3y: 95.9% 5y, 10y: 88%	3y, 5y: 96.1% 10y: 78.9%	N.M
11	Mattke et al., 2018 [21] Moderate risk	101 (54)	P.O carbon RT (n = 79) P.O proton RT (n = 22)	P.O carbon RT PR in 75 (95) Biopsy in 4 (5) P.O proton RT: PR in 18 (82) Biopsy in 4 (18)	Low grade, n = 101	Mean carbon RT boost volume = 34.9 cm^3^ Mean proton RT boost volume = 38.2 cm^3^	Carbon: 60 GyE Proton: 70 GyE	Carbon (median): 43.7 months Proton (median): 30.7 months	Carbon 1y: 98.6% 2y: 97.2% 4y: 90.5% Proton 1y, 2y,4y: 100%	Carbon 1y: 100% 3y: 98.5% 4y: 92.9% Proton: 1y, 3y, 4y: 100%	N.M
12	Hug et al., 1999 [22] Moderate risk	25 (9)	P.O proton RT (n = 25)	N.M	Low grade, n = 25	N.M	Mean: 70.7 GyE	Mean: 33.2 months	3y: 94% 5y: 75%	3y, 5y: 100%	N.M
13	Rosenberg et al., 1999 [23] Critical risk	200 (67)	P.O photon and proton RT (n = 200)	GTR in 10 (5) STR in 148 (74) PR in 42 (21)	Low grade, n = 200	N.M	Median: 72.1 GyE	Mean: 65.3 months	5y: 99% 10y: 98%	N.M	5y, 10y: 99%
14	Noel et al., 2001 [24] Moderate risk	11 (5)	P.O proton RT (n = 11)	N.M	Low grade, n = 11	Mean = 28 cm^3^	Photon median: 45Gy Proton median: 22 GyE Overall Mean: 66.7 GyCGE	Mean: 30.5 months	2y, 3y: 90%	3y: 90% 4y: 60%	N.M
15	Sahgal et al., 2014 [25] Serious risk	18 (10)	IG-IMRT photon RT (n = 18)	GTR in 7 (39) STR in 9 (50) Biopsy in 2 (11)	Low grade, n = 18	Mean = 24.6 cm^3^	Median: 70 GyE	Median: 66.5 months	5y: 88.1%	3y: 87.8% 5y: 84.1%	N.M
16	Feuvret et al., 2016 [26] Moderate risk	159 (72)	P.O proton RT or proton and photon RT (n = 159)	Complete in 13 (8) Incomplete in 133 (84) Biopsy 13 (8)	Low grade, n = 159	Mean = 23.1 cm^3^	Meidan total dose: 70.2 GyE Median proton: 36.6 GyE Median photon: 34.2 Gy	Clinical: 77 months Radiologic: 65 months	5y: 93.2% 10y: 84.2%	5y: 94.9% 10y: 87%	N.M
17	Weber et al., 2016 [27] Moderate risk	71 (31)	P.O proton RT (n = 71)	GTR in 3 (4) STR in 68 (96)	Low grade, n = 71	Mean = 36.1 cm^3^	Mean: 72.5 GyE	Mean: 50 months	5, 7y: 93.6%	N.M	N.M
18	Weber et al., 2016 [28] Moderate risk	77 (35)	P.O proton RT (n = 77)	N.M	Low grade, n = 73 High grade, n = 4	Median = 25.9 cm^3^	Median: 70 GyE	Mean: 69.2 months	5y: 94.2% 8y: 89.7%	5y, 8y: 93.5%	N.M
19	Weber et al., 2018 [29] Moderate to serious risk	251 (109)	P.O. proton (n=116) P.O proton ± photons (n = 135)	N.M.	N.M.	P.O. proton = 35.53 cm^3^ P.O proton ± photons = 22.22cm^3^	P.O. proton = 69.67 GyE P.O proton ± photons = 69.86 GyE	All patients (median): 87.3 months Surviving patients (median): 88.0 months	7y: 93.1%	7y: 93.6%	N.M
20	Holtzman et al., 2019 [30] Critical risk	43 (18)	P.O proton RT (n = 43)	N.M	Low grade, n =41 High grade, n = 2	Mean = 18 cm^3^	Median: 73.8 GyE	Median: 44.4 months	2y: 100% 3y: 93% 4y: 89%	2y: 98% 3y, 4y: 95%	2y, 3y, 4y: 100%
21	Kawashima et al., 2022 [31] Serious bias	51 (25)	Surgery with postoperative GKRS (n = 30) Surgery + GKRS as salvage therapy (n = 21)	GTR in 1 (2) STR in 18 (35) PR in 24 (47)	Low grade, n = 31 High grade, n = 2 Dedifferentiated subtype of nonconventional SBC, n = 1 Unknown, n = 17	Median: 8 cm^3^	Median: 16 Gy	Median: 62 months	3y: 87% 5y: 78% 10y: 67%	N.M	After GKRS: 3y, 5y: 96% 10y: 83% After disease diagnosis: 3y: 98% 5y, 15y: 90%
22	Pattankar et al., 2022 [32] Critical risk	7 (1)	P.O adjuvant GKRS (n = 7)	STR in 6 (86) PR in 1 (14)	N.M	Mean: 4.16 cm^3^	Mean: 18.29 Gy	Mean: 72 months Median: 60 months	5y: 57.1%	5y: 100%	N.M

This table shows the raw data of 22 included studies. It shows per study the number of patients included, the intervention used, the extent of resection (stated as gross-total resection, subtotal resection, partial resection, or biopsy), grade of the chondrosarcoma (low or high grade), mean/median tumor size (in cm^3^), mean/median dose (in GyE/CGE), and mean/median follow-up time (in months). Furthermore, it also shows the progression-free survival, overall survival, and disease-specific survival reported in each study. When PFS was not available, local control was used instead, and when that was not available, recurrence-free survival was used. GTR: gross tumor resection, STR: subtotal resection, NTR: near-total resection PR: partial resection, P.O. = postoperative, R.T = Radiotherapy, N.M: not mentioned, N/A = not applicable, y = years, n = number of patients.

**Table 3 cancers-16-00856-t003:** Reported recurrences, salvage treatment(s), and prognostic factors for tumor control and/or overall survival. P.O. = postoperative. N.M = not mentioned. LC = local control. OS = overall survival. FFS = failure-free survival.

Author, Year	Intervention	Number of Recurrences	Time to Recurrence (Months). Given as Mean/Median (Range)	Salvage Treatment	Risk Factors
Tzortzidis et al., 2006 [11]	Surgery ± radiotherapy	12 (52%)	Mean: 86 (2–255)	N.M	N.M
Simon et al., 2018 [12]	Surgery only P.O. Proton RT	Surgery only: 8 (33.3%) P.O proton RT: 1 (4.0%)	Mean: 51 (9–142)	Surgery in eight patients. Secondary proton therapy in five patients.	N.M
Hasegawa et al., 2021 [13]	Surgery only P.O. radiotherapy therapy (proton, GKRS)	Surgery only: 7 (39%) P.O. RT: 1 (7.0%)	Median: 38 (16–225) in surgery-only group Median: 33 in surgery + RT group	Surgery only: Radiotherapy alone in three patients. Surgery alone in three patients. Surgery + radiotherapy in one patient. P.O. radiotherapy: One patient treated with proton RT.	N.M
Vaz-Guimareaes et al., 2017 [15]	Surgery only	7 (20.0%)	23.5 (14–58)	Three underwent surgery and P.O. proton therapy. Two underwent developed lung metastasis and underwent chemotherapy. Two were not reported.	N.M
Hasegawa et al., 2018 [16]	Surgery only	1 (5.3%)	12	Surgery	N.M
Al-Shaibibi et al., 2023 [17]	Surgery only	3 (30.0%)	Mean: 101	One patient treated with salvage GK twice. One patient treated with proton therapy. No adjuvant therapy for third patient.	N.M
Liu et al., 2023 [18]	Surgery only	7 (26.9%)	N.M	One patient underwent surgery and radiotherapy. Two patients underwent—surgery alone. Three patients underwent radiotherapy alone.	OS: limited tumor excision, large tumor volume, and high pathological grading.
Schulz-Ertner et al., 2007 [19]	P.O. carbon-ion therapy	2 (3.7%)	Patient 1: 36 Patient 2: 48	Patient 1: photon radiation therapy (45 Gy). Patient 2: partial resection and postoperative reirradiation with carbon-ion radiation therapy (60 CGE).	N.M
Uhl et al., 2014 [20]	P.O. carbon-ion therapy	10 (12.6%)	Median: 91 (3–175)	N.M	LC: older age, high GTV
Mattke et al., 2018 [21]	Proton therapy Carbon-ion therapy	5 (5.0%)	Median: 29.8 (8.1–47.3)	N.M	LC: older age
Hug et al., 1999 [22]	P.O. proton therapy	2 (8.0%)	Mean: 33	N.M	LC: large tumor volume, brainstem involvement
Rosenberg et al., 1999 [23]	P.O. proton therapy	3 (1.5%)	Mean: 65.3 (2.1–222.2)	N.M	N.M
Noel et al., 2001 [24]	P.O. proton and photon therapy	2 (18.2%)	N.M	N.M	LC + OS: older age, high number of surgical resections prior, prolonged time to SRS
Sahgal et al., 2014 [25]	P.O. IGMRT Photon therapy	2 (11.0%)	Median: 15.6 (14.4–16.8)	N.M	LC: older age
Feuvret et al., 2016 [26]	P.O. proton therapy	6 (3.8%)	Median: 39.1	N.M	LC/OS/PFS: high age, large GTV, primary disease status, uncontrolled tumor status
Weber et al., 2016 [27]	P.O. proton therapy	5 (7.0%)	Mean: 50 (4–176)	N.M	N.M
Weber et al., 2016 [28]	P.O. proton therapy	6 (7.8%)	Median: 28.4 (11.7–140.8)	N.M	LC: large tumor volume, brainstem and/or optic apparatus compression, older age
Weber et al., 2018 [29]	Proton +/− photon radiotherapy	15 (5.9%)	Median: 43.6 (5.2–140.8)	N.M	FFS: large tumor volume, optic pathway compression OS: large tumor volume, older age, increased number of surgeries, brainstem/optic apparatus compression
Holtzman et al., 2019 [30]	P.O. proton therapy	3 (6.9%)	Median: 44.4	Surgery only in all patients.	N.M
Kawashima et al., 2022 [31]	P.O. GKRS Surgery only with GKRS as salvage	Progression in 12 (23.5%), of which 7 (13.7%) recurrences	Median time between initial GKRS and additional treatment: 47	Two patients —repeated GKRS. Three patients—surgery only (transcranial/endoscopic). Six patients—surgery + RT (incl GKRS). One patient—conservative management due to age.	LC: repeated recurrences, prescription doses ≥ 15 Gy and 14 Gy. Multivariate analysis showed higher prescription doses and no history of repeated recurrences were significant factors for better local control.
Pattankar et al., 2022 [32]	P.O. adjuvant GKRS	3 (42.9%)	Mean: 32.76	Two patients—surgery alone. One patient—surgery+ conventional EBRT.	N.M

This table reports the recurrences and their prognostic factors. It depicts the number of recurrences, mean/median follow-up to recurrence (in months), salvage therapy used, and the risk factors.

**Table 4 cancers-16-00856-t004:** Cumulated surgical complications.

Complication	No. of Patients (%)
Cranial nerve palsy	68 (45.0%)
Subcutaneous hydrops	4 (2.6%)
Dysphagia	3 (1.9%)
Facial numbness	2 (1.3%)
Cerebrospinal fluid leak	23 (15.2%)
Hearing loss	15 (9.9%)
Meningitis	8 (5.3%)
PE/DVT	5 (3.3%)
Cerebral ischemia	4 (2.6%)
Diabetes insipidus	3 (1.9%)
Cerebrovascular events	2 (1.3%)
ICA damage	2 (1.3%)
Hydrocephalus	2 (1.3%)
Miscellaneous	10 (6.6%)

This table shows the complications faced postoperatively reported in the 22 studies. Complications that had a frequency of less than 1% were grouped into the category ‘Miscellaneous’. Examples of complications that fell into this category included ataxia, temporal epilepsy, seizure, meningitis, and cerebral abscess.

**Table 5 cancers-16-00856-t005:** Cumulated acute radiation-induced adverse events. Grading as per CTCAE v4.0 [33].

Complication	Grade	No. of Patients (%)
Alopecia	1–2	202 (16.8%)
Middle ear effusion	1–2	159 (13.2%)
Asthenia	1–2	159 (13.2%)
Erythema	1–2	159 (13.2%)
Nausea/vomiting	1–2	159 (13.2%)
Fatigue	1–2	43 (3.5%)
Radiation dermatitis	1–2	43 (3.5%)
Mucositis	1–3	46 (3.8%)
Erythema	N.M	36 (2.9%)
Headaches	N.M	36 (2.9%)
Alopecia	N.M	32 (2.7%)
Loss of appetite	N.M	25 (2.1%)
Fatigue	N.M	25 (2.1%)
Nausea/vomiting	N.M	25 (2.1%)
Dysgeusia	N.M	15 (1.2%)
Xerostomia	N.M	15 (1.2%)
Mucositis	N.M	15 (1.2%)
External/middle ear otitis	N.M	11 (0.9%)
Acute parotitis	N.M	1 (0.08%)

This table shows the total acute radiation-induced adverse events reported in the 22 studies. The definition used for ‘acute’ varies in the 22 studies, and most do not define it. The complications are defined as per the common terminology criteria for adverse events (CTCAE) v4.0 [33]. When not reported, it is reported as not mentioned, ‘N.M’.

**Table 6 cancers-16-00856-t006:** Cumulated late radiation-induced adverse events. Grading as per CTCAE v4.0 [33].

Characteristic	Grade	No. of Patients (%)
Pituitary dysfunction	1–2	97 (38.8%)
Hearing loss	3–4	29 (11.6%)
Hearing loss	N.M	21 (8.4%)
Unspecified Grade 3+	3+	14 (5.6%)
Brain necrosis	3+	9 (3.6%)
Temporal lobe necrosis/lesions	1–2	9 (3.6%)
Hearing loss	1–2	5 (2.0%)
Bone necrosis	3	5 (2.0%)
Hypopituarism	N.M	5 (2.0%)
Temporal lobe necrosis/lesions	N.M	5 (2.0%)
Dizziness	N.M	4 (1.6%)
Memory loss	2	4 (1.6%)
Visual field deficits	N.M	4 (1.6%)
Spinal cord necrosis	3	3 (1.2%)
CN palsy	1–2	3 (1.2%)
CN palsy	N.M	3 (1.2%)
Miscellaneous		30 (12.0%)

This table shows the total late radiation-induced adverse events reported in the 22 studies. The grade of the complication is also mentioned, as per the common terminology criteria for adverse events (CTCAE) v4.0, when reported. When not reported, it is reported as not mentioned, ‘N.M’. Complications with a frequency of under 1% were all grouped into the ‘Miscellaneous’ category; examples of complications that fell under this category include cerebellum necrosis, spinal cord necrosis, stroke, hyperprolactinemia, serous otitis, and temporal lobe edema.

## Data Availability

No new data were created or analyzed in this study. Data sharing does not apply to this article.

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
