# Peer review of "Skull-Base Chondrosarcoma: A Systematic Review of the Role of Postoperative Radiotherapy"

_cancers, 2024, doi:10.3390/cancers16050856_

Round 1
Reviewer 1 Report
Comments and Suggestions for Authors
Dear editor and author,
This is a non-well-prepared review that aims to systematically map the research on Skull Base Chondrosarcoma, gather the latest available evidence regarding the treatment of skull base chondrosarcoma and analyze the long-term prognosis to ascertain the potential added value of adjuvant radiotherapy and its application. It presents a compromised method and inadequate processing of the data for a definite outcome. Therefore, I may reconsider it after major revision.
Comments 1:
The title is too general and not specific enough, more like the name of a monograph.
Comments 2:
In the Introduction part, partial references are missing from line 76-line 82.
Comments 3:
If this is a systematic review, the author should also register online in Prospero or another official website opening for the following researchers around this topic.
Comments 4:
There is no risk bias assessment in this systematic review which is considered as an essential part.
Comments 5: There is no meta-analysis in this review. Could the author add relevant tables or figures to illuminate the data analysis (Overall, over the entire cohort, progression free survival was estimated to be 98.1%, 91.0%, 90.3%, and 84.5%, at 1-, 3-, 5-, and 10-years respectively. Similarly, overall survival was estimated to be 100%, 95.9%, 94.2%, and 85.0% at the same timepoints)?
Comments 6:
The authors may re-check the discussion as most of the paragraphs are not supported by evidence (such as lines 626-641 without any references).
Comments 7:
The authors do not discuss the innovations and limitations of the review in sufficient detail.
Comments 8:
The conclusions are too brief as a systematic review.
Comments on the Quality of English LanguageDear editor and author,
This is a non-well-prepared review that aims to systematically map the research on Skull Base Chondrosarcoma, gather the latest available evidence regarding the treatment of skull base chondrosarcoma and analyze the long-term prognosis to ascertain the potential added value of adjuvant radiotherapy and its application. It presents a compromised method and inadequate processing of the data for a definite outcome. Therefore, I may reconsider it after major revision.
Comments 1:
The title is too general and not specific enough, more like the name of a monograph.
Comments 2:
In the Introduction part, partial references are missing from line 76-line 82.
Comments 3:
If this is a systematic review, the author should also register online in Prospero or another official website opening for the following researchers around this topic.
Comments 4:
There is no risk bias assessment in this systematic review which is considered as an essential part.
Comments 5: There is no meta-analysis in this review. Could the author add relevant tables or figures to illuminate the data analysis (Overall, over the entire cohort, progression-free survival was estimated to be 98.1%, 91.0%, 90.3%, and 84.5%, at 1-, 3-, 5-, and 10-years respectively. Similarly, overall survival was estimated to be 100%, 95.9%, 94.2%, and 85.0% at the same time points)?
Comments 6:
The authors may re-check the discussion as most of the paragraphs are not supported by evidence (such as lines 626-641 without any references).
Comments 7:
The authors do not discuss the innovations and limitations of the review in sufficient detail.
Comments 8:
The conclusions are too brief as a systematic review.
Author Response
Dear Reviewer,
Thank you for your valuable feedback. Your critical points will be taken into consideration to improve the manuscript. Cancers asks for the major revisions to be completed by 10 days, however this is too soon for us. It would be great if you could give us 6 weeks time to submit an improved manuscript integrating your suggested revisions. Once again, we thank you for your feedback and hope that you accept our new and improved manuscript.
Find attached a document, containing our preliminary point-by-point replies to your comments.
Kind regards,
Pawan Kishore Ravindran

Reviewer 2 Report
Comments and Suggestions for Authors
Authors present a systematic review on chondrosarcomas of the skull base. Authors have analyzed 18 articles; a total of 1250 patients were included in this cohort, of which 160 received surgery only. Overall survival was estimated to be 100%, 95.9%, 94.2%, and 85.0% (1,3,5,10 years) When analyzing the surgery-only group versus surgery and adjuvant radiotherapy group independently, the latter showed better long-term prognostic result; , higher age, brainstem/optic apparatus compression, and larger tumor volume prior to radiotherapy were found to be significant factors for local recurrence.
The study was not registered in PROSPERO database of systemic reviews. Part "overview of the studies" is a mere description of all studies involved in the manuscript, which is then being repeated in the table; one of both is not necessary. Statistical analysis is merely descriptive; we lack in the abstract statistical data and methods which support conclusion - which tests were used to determine that the surgery + radiotherapy was better than surgery alone. Surgical therapy was not thoroughly discussed - in these types of tumors, gross total resection with reconstruction of the skull base plays a crucial role in avoiding recurrence.
Several important references especially concerning particle therapy have been left out:
Takahashi M, Mizumoto M, Oshiro Y, Kino H, Akutsu H, Nakai K, Sumiya T, Ishikawa E, Maruo K, Sakurai H. Risk Factors for Radiation Necrosis and Local Recurrence after Proton Beam Therapy for Skull Base Chordoma or Chondrosarcoma. Cancers (Basel). 2023 Dec 1;15(23):5687. doi: 10.3390/cancers15235687. PMID: 38067389; PMCID: PMC10705337. Holtzman AL, Seidensaal K, Iannalfi A, Kim KH, Koto M, Yang WC, Shiau CY, Mahajan A, Ahmed SK, Trifiletti DM, Peterson JL, Koffler DM, Vallow LA, Hoppe BS, Rutenberg MS. Carbon Ion Radiotherapy: An Evidence-Based Review and Summary Recommendations of Clinical Outcomes for Skull-Base Chordomas and Chondrosarcomas. Cancers (Basel). 2023 Oct 17;15(20):5021. doi: 10.3390/cancers15205021. PMID: 37894388; PMCID: PMC10605639. Sallabanda M, Vera JA, Pérez JM, Matute R, Montero M, de Pablo A, Cerrón F, Valero M, Castro J, Mazal A, Miralbell R. Five-Fraction Proton Therapy for the Treatment of Skull Base Chordomas and Chondrosarcomas: Early Results of a Prospective Series and Description of a Clinical Trial. Cancers (Basel). 2023 Nov 25;15(23):5579. doi: 10.3390/cancers15235579. PMID: 38067283; PMCID: PMC10705113. Comments on the Quality of English LanguageAcceptable.
Author Response
Dear Reviewer,
Thank you for your valuable feedback. Your critical points will be taken into consideration to improve the manuscript. Cancers asks for the major revisions to be completed by 10 days, however this is too soon for us. It would be great if you could give us 6 weeks time to submit an improved manuscript integrating your suggested revisions. Once again, we thank you for your feedback and hope that you accept our improved manuscript.
Find attached a document containing our preliminary point-by-point replies to your comments.
Kind regards,
Pawan Kishore Ravindran

Round 2
Reviewer 2 Report
Comments and Suggestions for Authors
Authors have sufficiently responded to reviewer remarks.
Comments on the Quality of English LanguageAcceptable.
Author Response
Dear Reviewer,
Thank you for reviewing the paper! Glad to see that we have sufficiently responded to the remarks.
Kind regards,
Pawan Kishore Ravindran